# Otilonium bromide boosts antimicrobial activities of colistin against Gram-negative pathogens and their persisters

Chen Xu[1], Chenyu Liu[1], Kaichao Chen[1], Ping Zeng[2], Edward Wai Chi Chan[1,2] & Sheng Chen [1✉]

Colistin is the last-line antibiotic against Gram-negative pathogens. Here we identify an FDA-approved drug, Otilonium bromide (Ob), which restores the activity of colistin against colistin-resistant Gram-negative bacteria in vitro and in a mouse infection model. Ob also reduces the colistin dosage required for effective treatment of infections caused by colistin-susceptible bacteria, thereby reducing the toxicity of the drug regimen. Furthermore, Ob acts synergistically with colistin in eradicating multidrug-tolerant persisters of Gram-negative bacteria in vitro. Functional studies and microscopy assays confirm that the synergistic antimicrobial effect exhibited by the Ob and colistin involves permeabilizing the bacterial cell membrane, dissipating proton motive force and suppressing efflux pumps, resulting in membrane damages, cytosol leakage and eventually bacterial cell death. Our findings suggest that Ob is a colistin adjuvant which can restore the clinical value of colistin in combating life-threatening, multidrug resistant Gram-negative pathogens.

[1] Department of Infectious Diseases and Public Health, Jockey Club College of Veterinary Medicine and Life Sciences, City University of Hong Kong, Kowloon, Hong Kong. [2] State Key Lab of Chemical Biology and Drug Discovery, Department of Applied Biology and Chemical Technology, The Hong Kong Polytechnic University, Hung Hom, Kowloon, Hong Kong. ✉email: shechen@cityu.edu.hk

Infectious diseases caused by multidrug-resistant (MDR) pathogens have become a serious and rapidly worsening public health problem worldwide[1]. According to a previous review on antimicrobial resistance, infections caused by MDR strains may cause over 10 million deaths per year by 2050 if no proactive measures are taken to slow down the rapidly increasing trend of drug resistance[2]. The emergence of the resistance determinants $bla_{NDM-1}$ and $bla_{KPC-2}$, which are highly transmissible among strains of various species and known to encode carbapenem resistance, causes further reduction in the number of therapeutic options that remain[3–5]. Colistin, also called polymyxin E, is a polycationic antimicrobial peptide regarded as one of the last-resort antibiotics that can be used to treat infections caused by MDR Gram-negative strains, especially carbapenem-resistant Enterobacteriaceae (CRE), due to its high efficacy and low resistance rate[6]. Colistin exhibits electrostatic binding affinity to the negatively charged lipid A in the lipopolysaccharide (LPS) molecules located on the surface of Gram-negative bacteria, leading to the membrane damage and leakage of cellular contents, and hence bacterial cell death[7]. However, the clinical value of colistin is limited by its nephrotoxicity and neurotoxicity[8]. In addition, colistin resistance is increasingly being reported. Mutations in the phoPQ and pmrAB genes, which encode two different component systems, as well as the mgrB gene mutations, were reported to be associated with over-expression of lipid A modification enzymes such as EptA, as well as reduced affinity of Lipid A to polymyxins[9,10]. In recent years, a new mechanism of colistin resistance mediated by the plasmid-encoded enzyme MCR-1, which catalyzes the modification of lipid A, has been reported worldwide[11–16]. Worse still, the mobilizable colistin resistance gene mcr-1 has increasingly been acquired by CRE strains carrying the $bla_{KPC}$[17], $bla_{NDM}$[18], and $bla_{VIM}$ genes[19], rendering those strains able to evolve into real "superbugs" which are resistant to almost all known antimicrobial agents. Therefore, novel strategies that can help overcome colistin resistance in Gram-negative pathogens especially CRE are urgently needed. Compared with the time-consuming and expensive process of developing new antibiotics, identifying colistin adjuvants that can restore colistin activity and reduce the treatment dosage is a more effective and eco-friendly strategy to restore or even enhance the clinical value of colistin as a last-line antibiotic[20,21].

Otilonium bromide (Ob, IUPAC: N,Ndiethyl-N-methyl-2-[(4-benzoyl)oxy]ethanaminium), an FDA-approved antispasmodic drug, is extensively applied for treatment of patients suffering from irritable bowel syndrome (IBS)[22]. It has been shown that Ob could block the L- and T-type $Ca^{2+}$ channels and muscarinic and tachykinin receptors in the smooth muscle[23]. Ob is poorly absorbed by human cells so that there is no significant side effect reported for its clinical use[24]. Although the antimicrobial effects of Ob on Clostridioides difficile[25] and Staphylococcus aureus[26] have been reported in previous studies, the potential of using Ob for treatment of infections caused by Gram-negative pathogens has not been explored. In this study, we observed a strong synergistic antimicrobial effect when Ob and colistin were used together for inhibiting the growth of both colistin-resistant and susceptible Gram-negative strains in vitro. The drug combination was found to be particularly effective in treatment of infections caused by colistin-resistant CRE in in vivo experiments. The mechanisms underlying this synergistic antimicrobial effect were investigated in this work. The discovery of Ob as a new and safe colistin adjuvant provides a novel option for combating infections caused by MDR Gram-negative pathogens.

## Results and discussion

**Ob potentiates colistin activity against both colistin-susceptible and colistin-resistant bacteria in vitro.** In this study, E. coli J53 strain carrying a natural mcr-1-bearing IncI2 plasmid (33 kb, KX711706.1) was used to screen and identify colistin adjuvants which could enhance the antibacterial activity of colistin[27]. Using the checkerboard dilution assay, Ob was found to act synergistically with colistin (FICI = 0.25), conferring a 32-fold reduction in colistin MIC (from 8 μg/ml to 0.28 μg/ml) when 20 μg/ml of Ob was used in the susceptibility test. This degree of reduction in MIC is sufficient to bring the MIC of the majority of the test strains to a level below the clinical breakpoint (2 μg/ml, according to CLSI 2020) (Fig. 1a). Ob also acts synergistically with colistin against colistin-susceptible E. coli J53 strain (FICI ≤ 0.141), reducing the MIC from 1 μg/ml to ≤0.016 μg/ml when 10 μg/ml of Ob was used (Fig. 1b). We next tested if such synergistic antimicrobial effect was also observable in other Gram-negative strains and found that the colistin and Ob combination exhibited synergistic antimicrobial effect on both colistin-susceptible and colistin-resistant strains of various Gram-negative pathogens. This finding suggests that Ob is a potential broad-spectrum colistin adjuvant that can act on all major Gram-negative bacterial pathogens including

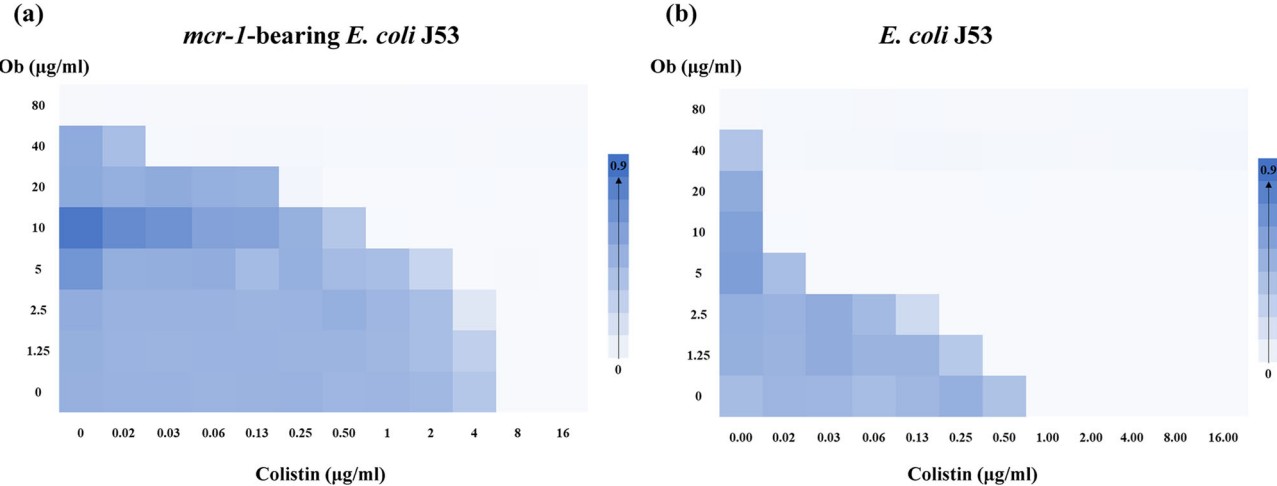

**Fig. 1 Checkerboard analysis of the synergistic antimicrobial effect of colistin and Ob on both colistin-resistant and colistin-susceptible E. coli.**
**a** Colistin-susceptible E. coli J53. **b** Colistin-resistant E. coli J53 carrying a mcr-1-bearing plasmid which was originally recovered from a clinical E. coli strain. CT, colistin.

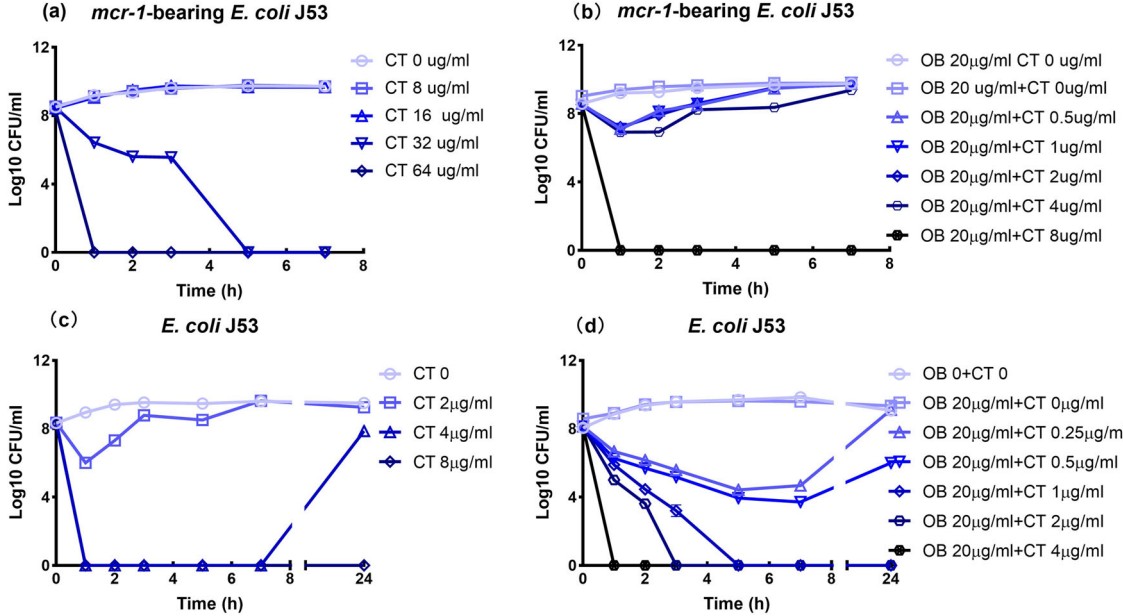

**Fig. 2 Time-kill assays of *E. coli* treated with colistin and colistin/Ob combination. a** The *mcr-1*-bearing *E. coli* J53 strain was treated with various concentrations of colistin alone and **b** in combination with 20 μg/ml Ob. **c** *E. coli* J53 was treated with various concentrations of colistin alone and **d** in combination with 20 μg/ml Ob. Data are mean ± SEM for *n* = 3 biologically independent experiment.

*Pseudomonas aeruginosa, Acinetobacter baumannii,* and *Salmonella* spp. (Supplementary Table 1).

To further evaluate the synergistic antimicrobial effect of Ob and colistin on colistin-resistant and colistin-susceptible *E. coli*, time-kill curves were constructed for the test strains growing in the exponential phase (Fig. 2). The growth of colistin-resistant *E. coli* could only be inhibited by colistin at 32 μg/ml and higher concentrations. However, the effective bactericidal concentration of colistin could be reduced to 8 μg/ml in the presence of 20 μg/ml of Ob. For colistin-susceptible *E. coli*, 8 μg/ml of colistin was required to eradicate the strains; in the presence of 20 μg/ml of Ob, however, 1 μg/ml colistin was sufficient to achieve complete eradication of the organism (Fig. 2). These data suggest that Ob indeed enhances the bactericidal activity of colistin against both colistin-resistant and colistin-susceptible *E. coli* dramatically, implying that the mechanism is not limited to inhibition of the colistin resistance mechanism.

The strong synergistic antimicrobial effect exhibited by Ob and colistin was visualized by the LIVE/DEAD cell viability assay. Fluorescence microscopy was performed to quantify the number of live and dead cells and determine the degree of reduction in colistin concentration in the drug combination required for effective eradication of the test organisms. Almost all *mcr-1*-bearing *E. coli* strains were alive upon treatment with saline, 8 μg/ml of colistin alone or 20 μg/ml of Ob alone. In the presence of 20 μg/ml of Ob, 4 μg/ml of colistin could eradicate >97% of organisms (Supplementary Fig. 1).

**Ob acts synergistically with colistin to eliminate clinical colistin-resistant *E. coli* strains in mouse infection model.** The in vivo antimicrobial effect of the Ob and colistin combination was further tested in the mouse sepsis model, with results showing that Ob could re-sensitize colistin-resistant CRE to colistin. Our data showed that all mice died within 36 h upon treatment with saline, colistin alone, or Ob alone. Treatment with a combination of Ob and colistin could successfully rescue 80% of the animals at 48 hpi, suggesting that Ob exhibits synergistic antimicrobial effect with colistin in treatment of infections caused by colistin-resistant CRE strains (Fig. 3).

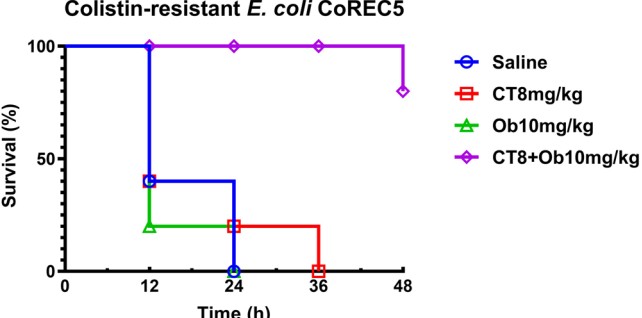

**Fig. 3 Assessment of antimicrobial efficacy of the Ob and colistin combination in mouse sepsis model infected with colistin-resistant *E. coli* CoREC59.** The mice were infected with ~$10^8$ CFU of bacteria and subjected to different treatments including saline, colistin alone (8 mg/kg), Ob alone (10 mg/kg), and a combination of colistin (8 mg/kg) and Ob (10 mg/kg). The number of dead mice was recorded during the experimental period. CT, colistin.

**Ob and colistin combination eliminates tolerant Gram-negative bacterial cells in vitro.** All bacterial populations, including those of non-antibiotic resistant strains, are known to harbor drug-tolerant sub-population that do not respond to antimicrobial actions of antibiotics. Such antibiotic tolerant sub-population are now known to be the culprit of a wide range of chronic and recurrent infections, especially among immunocompromised patients[28,29]. We therefore tested the bactericidal effect of the Ob and colistin combination on antibiotic tolerant bacterial sub-population. Ob was found to strongly enhance the efficacy of colistin in killing the starvation-induced bacterial tolerant cells of colistin-resistant and colistin-susceptible *E. coli* and *A. baumannii* strains. Compared to the initial population size of ~$10^8$ CFU/mL of *mcr-1*-bearing *E. coli* strain re-suspended in saline, the size of the viable population remained at a high level of $5 \times 10^5$ CFU/ml upon treatment with colistin at 32 μg/ml for 24 h. However, the entire drug-tolerant population was effectively eradicated by 2 μg/ml of colistin in the presence of 20 μg/ml of Ob in 24-h treatment. Ob could also act

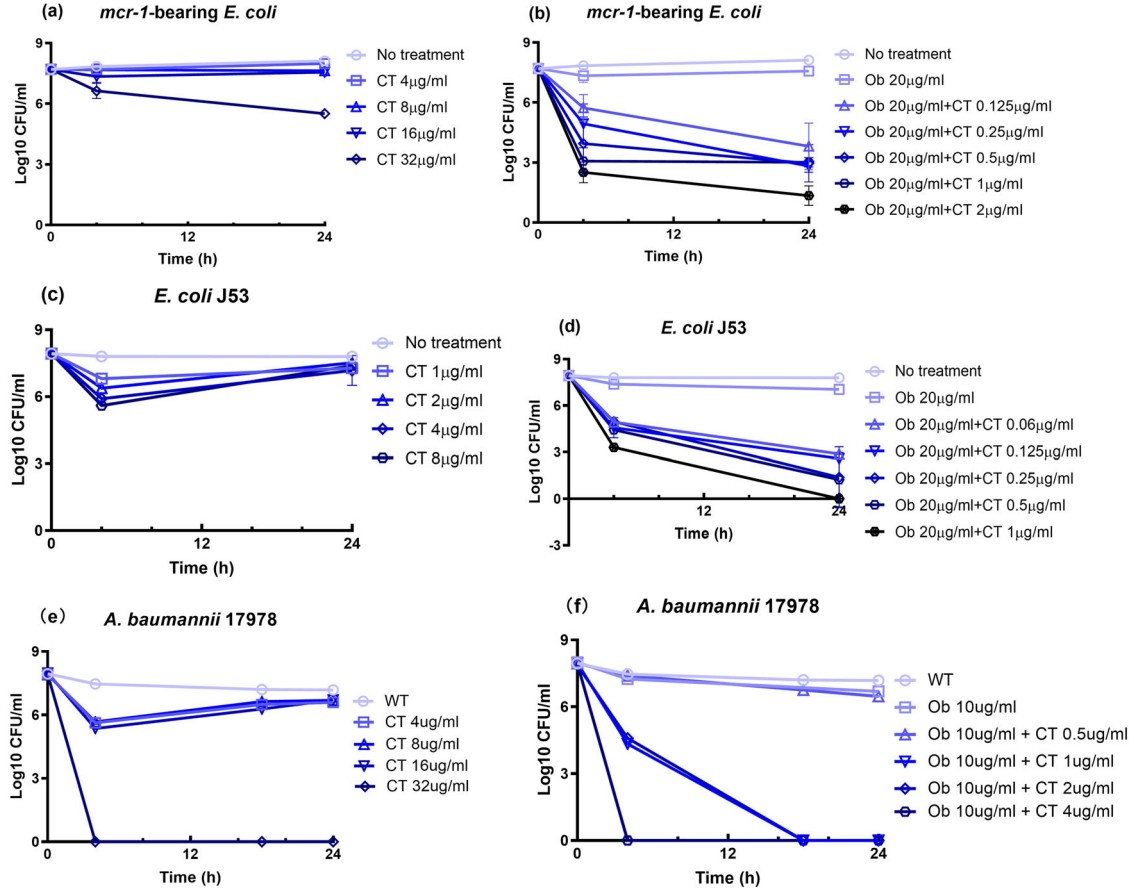

**Fig. 4 Assessment of the synergistic antimicrobial effect of Ob and colistin on starvation-induced persisters of colistin-resistant and colistin-susceptible ESKAPE bacterial pathogens.** All strains had been starved for 24 h prior to treatment with various concentrations of colistin alone, Ob alone or a combination of both. **a** Population size of *mcr-1*-bearing *E. coli* treated with different concentrations of colistin; **b** population size of *mcr-1*-bearing *E. coli* treated with different concentrations of colistin and 20 μg/ml Ob; **c** population size of colistin-susceptible *E. coli* J53 treated with different concentrations of colistin; **d** population size of colistin-susceptible *E. coli* J53 treated with different concentrations of colistin and 20 μg/ml Ob; **e** population size of colistin-susceptible *A. baumannii* 17978 treated with different concentrations of colistin; **f** population size of colistin-susceptible *A. baumannii* 17978 treated with different concentrations of colistin and 10 μg/ml Ob. CT, colistin. Data are mean ± SEM for *n* = 3 biologically independent experiment.

synergistically with colistin on the tolerant sub-population of colistin-susceptible *E. coli* and *A. baumannii* strains by reducing the concentration of colistin required to completely eradicate such sub-population to 1 μg/ml in a 24-h treatment (Fig. 4). These data suggest that Ob has high potential to be developed into a therapeutic agent for eradication of bacterial tolerant sub-population.

**Ob enhances the ability of colistin to cause membrane disruption.** Colistin exhibits high bactericidal efficacy against Gram-negative pathogens through specifically binding to the negatively charged phosphate group of lipid A in LPS in the cell membrane of Gram-negative bacteria, causing an increase in cell membrane permeability and leakage of cellular contents[30]. The mechanism of colistin resistance in Gram-negative bacteria mainly involves modification of lipid A, which renders colistin binding ineffective[31]. We therefore hypothesized that Ob might restore the ability of colistin to disrupt bacterial cell membrane. To test this hypothesis, we first visualized the morphological changes in colistin-resistant *E. coli* upon treatment with sub-MIC of colistin (8 μg/ml), Ob (20 μg/ml), and the combination of both, using SEM. No morphological changes were observed when *mcr-1*-bearing *E. coli* was treated with sub-MIC of Ob or colistin. When treated with 4 μg/ml of colistin and 20 μg/ml of Ob, the bacterial envelope was completely disrupted and leakage of the cytosol could be observed, along with shrinkage of the cell membrane.

Treatment with 20 μg/ml Ob, 8 μg/ml of colistin resulted in more severe disruption of cell membrane, suggesting that the presence of Ob could restore the killing effect of colistin on colistin-resistant *E. coli* (Fig. 5). To further confirm this theory, SYTOX Green staining analysis was employed to assess the membrane permeability of colistin-resistant *E. coli* before and after treatment with the colistin and Ob combination. SYTOX Green is a green-fluorescence nucleotide dye used to test membrane permeability and membrane integrity[32]. Green fluorescence can be detected by fluorescence spectrometer when bacterial cell integrity is destroyed and membrane permeability increases. Consistently, the data showed that Ob and colistin could each cause a significant increase in fluorescence in *mcr-1*-bearing *E. coli*, indicating that both drugs could cause increase in bacterial cell membrane permeability (Fig. 6a, b). The ability of Ob in enhancing membrane permeability was lower than colistin. As expected, treatment with the Ob and colistin combination caused a drastic increase in membrane permeability, suggesting that the effective bactericidal concentration of colistin can be reduced in the presence of Ob (Fig. 6c). Furthermore, microscopy imaging of *mcr-1*-bearing *E. coli* stained with SYTOX depicted a much more dramatic increase in fluorescence when the strain was treated with a combination of 20 μg/ml Ob and 8 μg/ml colistin, when compared to monotreatment with colistin or Ob at the same concentration (Fig. 6d).

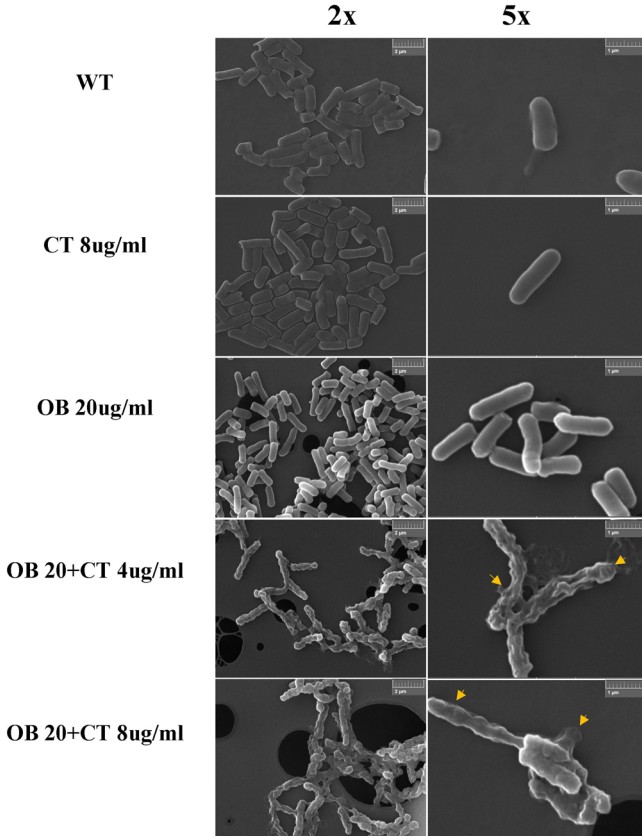

**Fig. 5 Scanning electronic microscopy of *mcr-1*-bearing *E. coli* J53 treated with colistin alone, Ob alone, and various combinations of colistin and Ob.** CT, colistin.

Previous studies demonstrated that electrostatic interaction between colistin and the negatively charged LPS caused displacement of divalent ions ($Ca^{2+}$ and $Mg^{2+}$) from the phospholipids and hence disruption of the bacterial cell membrane[33,34]. Divalent ions act as a cross-linker that allows networking of LPS molecules, so that the membrane becomes structurally stable, tightly packed, and only selectively permeable[35]. The effect of divalent ions on the antibacterial activity of Ob and colistin was further investigated. Addition of $Mg^{2+}$ and $Ca^{2+}$, particularly $Ca^{2+}$, could suppress the activity of colistin and the Ob and colistin combination on both colistin-resistant and susceptible *E. coli*. We also determined the activity of Ob and colistin on *P. aeruginosa*, with results showing that supplementation of calcium ions could also suppress the bactericidal effect of colistin and the drug combination on this pathogen (Supplementary Fig. 2).

**Ob acts as a membrane proton motive force dissipator (PMF).** Proton motive force (PMF; $\Delta P$) is an electrochemical gradient of protons generated by the electron transport chain in bacteria, which acts by extruding protons out of the cells. PMF, which is necessary for ATP synthesis and transport of various solutes[36], is the sum of two parameters: the electric or membrane potential ($\Delta\varphi$) and the transmembrane proton gradient ($\Delta pH$). To further investigate the nature of damage inflicted by Ob and colistin on bacterial cell membrane, the membrane potential of colistin-resistant *E. coli* was determined using 3,3-dipropylthiadicarbocyanine iodide (DiSC3(5)). Valinomycin, a $K^+$ transporter, was used as the positive control which can cause complete dissipation of bacterial membrane potential. Upon dissipation of membrane potential, DiSC3(5) cannot be anchored onto the bacterial membrane of *mcr-1*-bearing *E. coli* and would be released to the

medium, resulting in a marked increase in fluorescence intensity. Ob was found to act as a strong membrane potential dissipator which caused a PMF dissipation rate even higher than that of valinomycin. The combined use of Ob and colistin caused extremely rapid dissipation of PMF in colistin-resistant *E. coli* (Fig. 7a–d). Consistently, microscopy image of *mcr-1*-bearing *E. coli* could be stained with DiSC3(5), whereas treatment with 20 μg/ml Ob caused membrane potential dissipation and the release of the dye to the medium (Fig. 7e). We also investigated the effect of Ob and colistin on the membrane potential of colistin-susceptible *E. coli* J53. Ob itself could also cause dissipation of membrane potential of the colistin susceptible strains, indicating that the PMF dissipation effect of Ob does not involve neutralizing the effect of the *mcr-1* gene product (Supplementary Fig. 3).

**Ob and colistin combination suppresses efflux activities.** Since PMF is required for driving efflux activities, the effect of the combination of Ob and colistin on efflux activities was determined using the Nile Red efflux assay[37]. Nile red is a substrate of various efflux pumps which becomes strongly fluorescent upon partitioning into the bacterial membrane, and could be pumped out of the cell immediately[38]. Carbonyl cyanide m-chlorophenylhydrazone (CCCP), an PMF dissipator, was employed as a positive control to inactivate efflux pumps effectively[39]. Monotreatment with either 20 μg/ml Ob or 8 μg/ml colistin caused rapid reduction in Nile red export in *mcr-1*-bearing *E. coli* J53, with a reduction rate even higher than that recorded for 10 μM CCCP (Fig. 8a, b). The combined use of Ob and colistin exerted an even stronger inhibitory effect on Nile red efflux, confirming that Ob and colistin act synergistically in inhibiting Nile red efflux (Fig. 8c). Addition of 50 mM glucose could energized the efflux pump and trigger the efflux again, so that the inactivation effect of CCCP was effectively counteracted and Nile red could again be pumped out to the medium immediately. In comparison, the energization effect of glucose on de-energized cells subjected to treatment with Ob, colistin and the combination of both, was lower than that recorded in CCCP-treated cells. The synergistic effect of Ob and colistin on inactivation of Nile red efflux was also observed in colistin-susceptible *E. coli* J53, with results confirming that the inhibitory effect on efflux pump exerted by Ob and colistin does not involve interaction with the *mcr-1* gene product (Fig. 8d).

**Ob suppresses bacterial motility.** As a previous study reported that flagellar formation was dependent on PMF and that flagellar motor rotation was driven by PMF, we further determined the effect of Ob on bacterial motility[40]. *P. seruginosa* exhibited swarming motility on semisolid surface; such motility was dependent upon a functional flagella[41,42]. As shown in Fig. 9, the migration distance of PAO1 inoculated onto a semisolid agar plate containing Ob was found to decrease in a dose-dependent manner after overnight incubation. Gramicidin, an ionophore acting as PMF dissipator, was used as the positive control. Furthermore, a significant increase in migration distance of PAO1 treated with 40 μM Ob supplemented with 10 mM calcium ions was observed, suggesting that 10 mM calcium ions could effectively suppress the inhibitory effect of Ob on bacterial swarming.

It has been reported that the combination of colistin and adjuvant compounds that could restore colistin activity offered a promising strategy to treat infections caused by colistin-resistant pathogens[21]. Although previous studies have recommended Ob in treatment of IBS by using it as a calcium channel blocker[43], its potential in treating bacterial infections has not been extensively studied. Ob was shown to exhibit antimicrobial effect on *Staphylococcus aureus*, *Clostridium difficile*, and *A. baumannii*

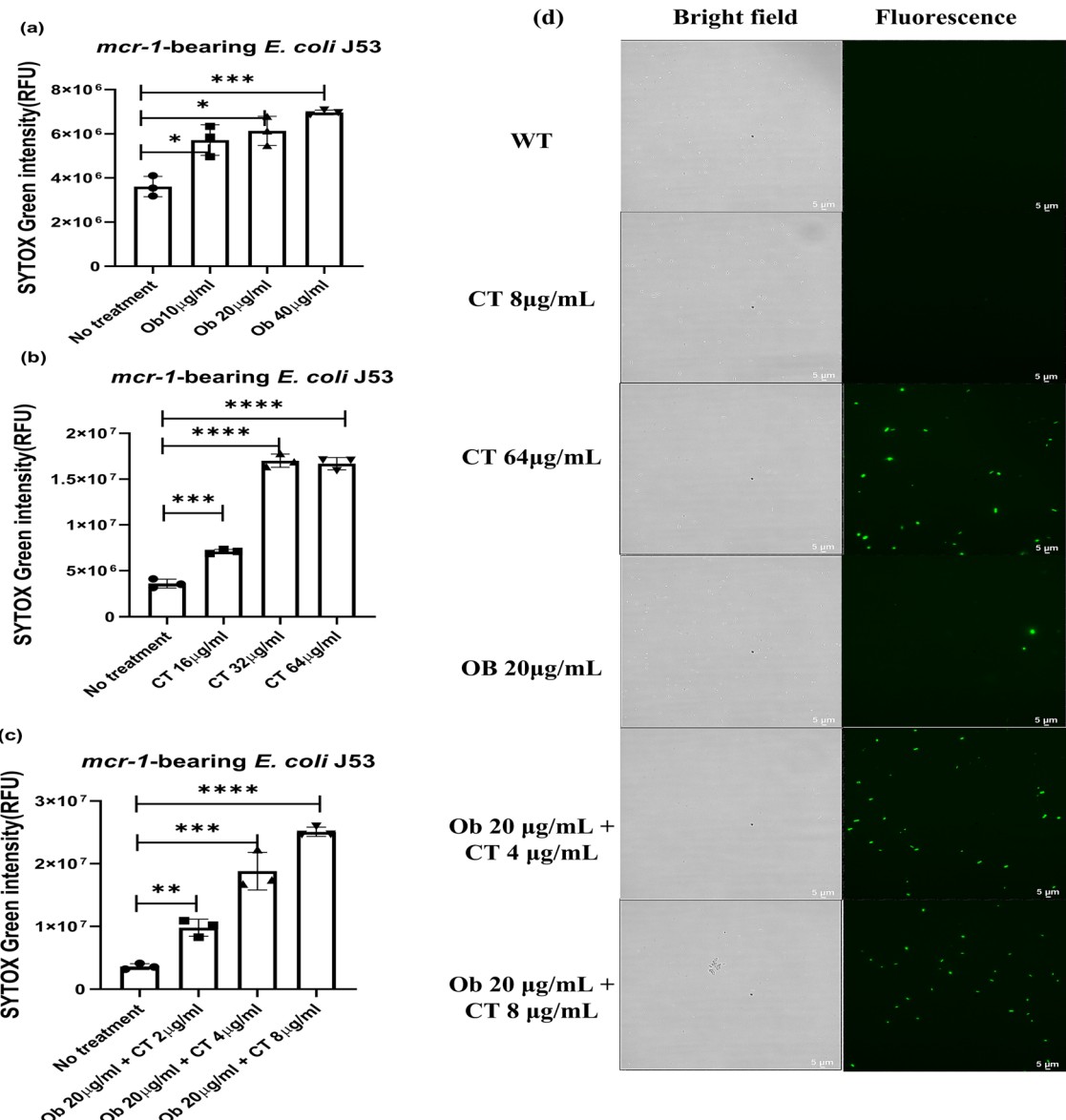

**Fig. 6 Ob enhances the membrane-permeabilizing effect of colistin on colistin-resistant _E. coli._** The bacterial membrane permeability of _mcr-1_-bearing _E. coli_ J53 treated with **a** Ob, **b** colistin, and **c** a combination of both agents was determined by using SYTOX green. **d** Microscopic images of bacterial cell membrane permeability of _mcr-1_-bearing _E. coli_ J53 treated with Ob, colistin, and a combination of both agents. Statistical analysis was performed using unpaired two-tailed student _t_-test. Data are mean ± SEM for $n = 3$ biologically independent experiment. *$p < 0.05$, **$p < 0.005$, ***$p < 0.0005$, ****$p < 0.00005$.

through causing membrane damages[26,44]. In this study, although Ob itself is only weakly bactericidal, it exhibits strong synergistic effect when used in combination with colistin in treatment of infection caused by both colistin-resistant and susceptible Gram-negative pathogens. Ob is an FDA-approved drug without major side effects in human[24], and can effectively reduce the treatment dosage of colistin and hence the toxicity of colistin on mammalian cells. It should be noted that the neurotoxicity and nephrotoxicity of colistin is the key factor limiting its clinical value[45].

In the mechanistic study, Ob was found to act synergistically with colistin to permeabilize bacterial cell membrane, dissipate PMF, inhibit multidrug efflux pump function and suppress bacterial motility. The increased membrane permeability induced by the combination of Ob and colistin allows for self-promoted uptake of the colistin molecule to further enhance its damaging effect[46]. The action of Ob and colistin could both be suppressed by high concentration of divalent ions (especially $Ca^{2+}$ ions), suggesting that they could each displace the membrane-stabilizing divalent ions, resulting in an increase in membrane permeability. Furthermore, Ob is an amphipathic compound that contains both a hydrophilic quaternary amine and a hydrophobic long carbon chain. These functional groups enable it to partition into the phospholipid bilayer of the membrane and cause membrane damage[47,48].

The displacement of the membrane-stabilizing magnesium and calcium ions by Ob and colistin also results in disruption of the electron transport chain and hence further dissipation of proton motive force. Ob could also act as a calcium channel blocker to inhibit the transportation of the membrane-stabilizing calcium ions. Dissipation of PMF upon treatment by Ob and colistin is associated with membrane depolarization and dysfunction, as well as metabolic perturbation. The metabolic status of the bacterial cell is known to be severely impacted by the actions of

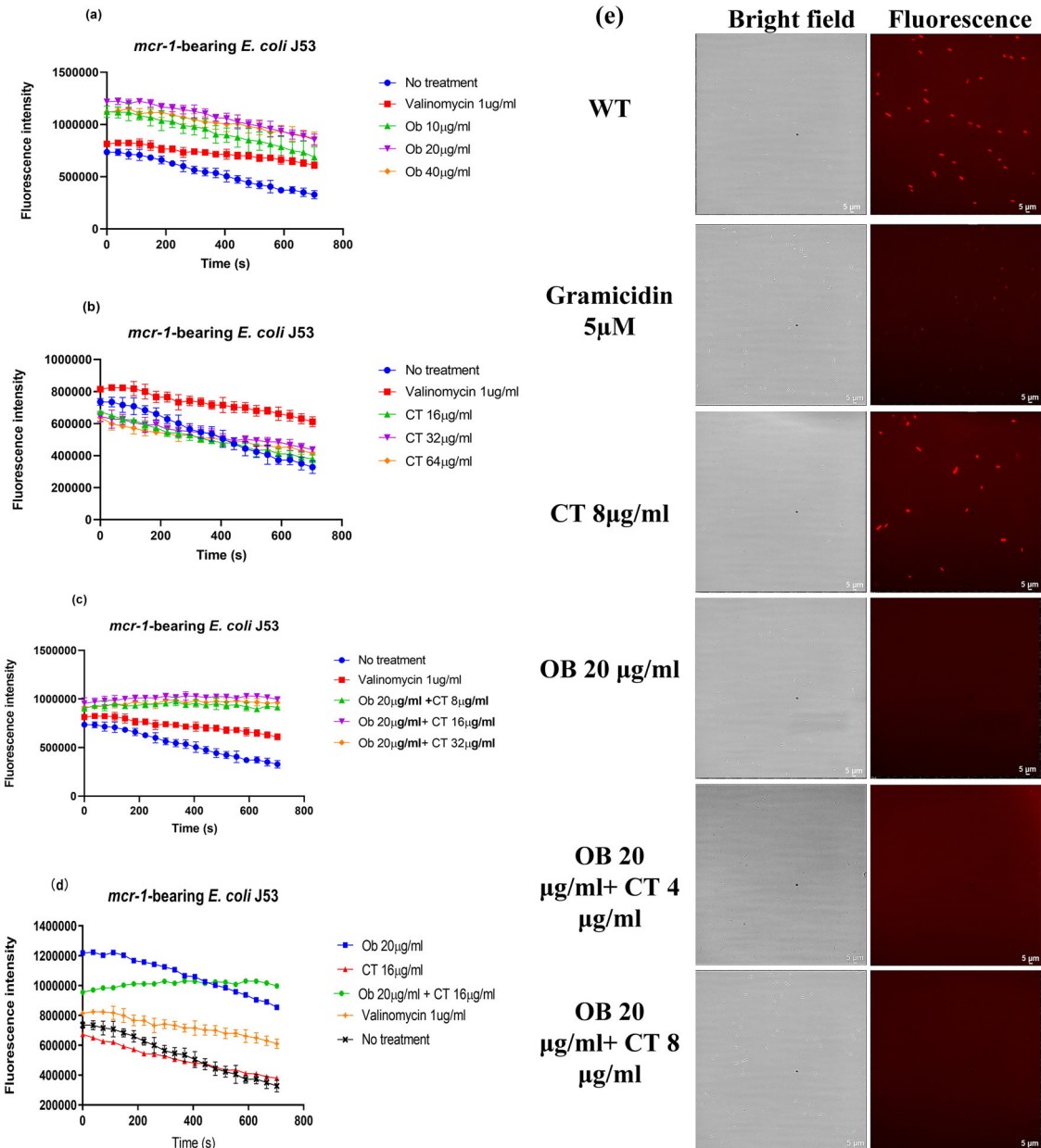

**Fig. 7 Ob dissipates the bacterial membrane potential.** Dissipation of membrane potential in *mcr-1*-bearing *E. coli* J53 upon treatment with Ob (**a**), colistin (**b**), and various combinations of the two drugs (**c**). **d** Comparison of effects of different compounds on the bacterial cell membrane potential of *mcr-1*-bearing *E. coli* J53. Valinomycin, a K$^+$ ionophore, was used as positive control. The assay was conducted in the presence of 100 mM of KCl. CT, colistin. **e** Fluorescence microscopic images of bacterial membrane potential of *mcr-1*-bearing *E. coli* J53 taken upon treatment with Ob and colistin. Gramicidin, an ionophore, was used as the positive control. Data are mean ± SEM for $n = 3$ biologically independent experiment.

antimicrobial agents[49]. Since PMF is required to drive efflux activities, PMF dissipation results in suppression of efflux and rapid accumulation of drugs and hence further cellular damages. Suppression of bacterial motility also leads to increased exposure of bacterial cells to Ob and colistin, thereby enhancing their antimicrobial efficacy (Fig. 10)[50].

**Conclusions**. The clinical value of colistin has been undermined by its cytotoxicity and the emergence of resistant strains. Development of colistin adjuvant is the most effective strategy to overcome the public health threat associated with the emergence of colistin-resistant Gram-negative pathogens, especially the colistin-resistant CRE strains. In this study, we identified Ob as a promising colistin adjuvant that can restore the bactericidal

effect of colistin by allowing it to eradicate colistin-resistant Gram-negative strains both in vitro and in mouse infection model. The activity of colistin against colistin-susceptible Gram-negative strains can also be enhanced by Ob, so that a much smaller amount of the drug can be used in clinical treatment. Furthermore, Ob exhibits synergistic bactericidal effect with colistin when tested on starvation-induced tolerant Gram-negative bacterial cells in vitro. The Ob and colistin combination were found to cause membrane permeabilization, dissipation of PMF, and inhibition of efflux activities, leading to extensive membrane damage and eventually cell death. Based on these observations, we believe that Ob, which is already a FDA-approved drug, is a reliable colistin adjuvant which can fully restore and even enhance the clinical value of colistin. This drug

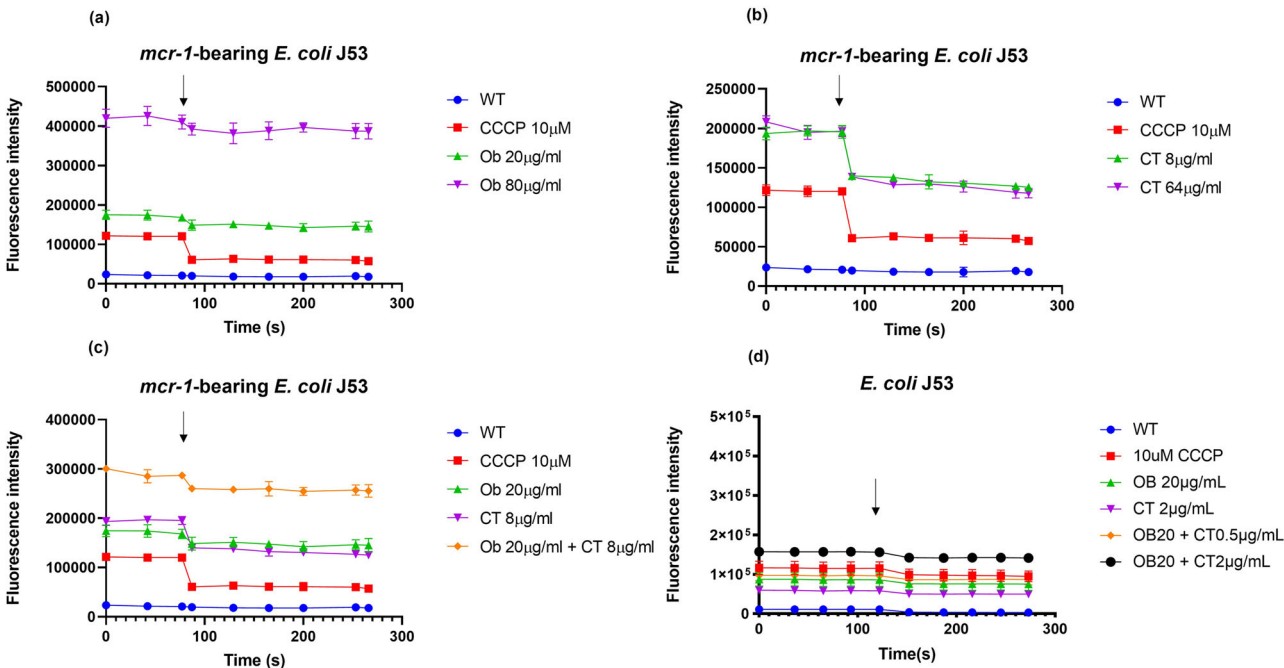

**Fig. 8 Ob and colistin suppress efflux activities.** Inhibition of Nile Red efflux in *mcr-1*-bearing *E. coli* by Ob (**a**), colistin (**b**), and a combination of both agents (**c**). 50 mM of Glucose was added to the assay at 77 s and fluorescence was monitored for 3 min. **d** Nile Red efflux of colistin-susceptible *E. coli* J53 treated with Ob, colistin, and a combination of both agents. 50 mM of Glucose was added to the assay at 110 s and fluorescence was monitored for 3 min. Data are mean ± SEM for *n* = 3 biologically independent experiment.

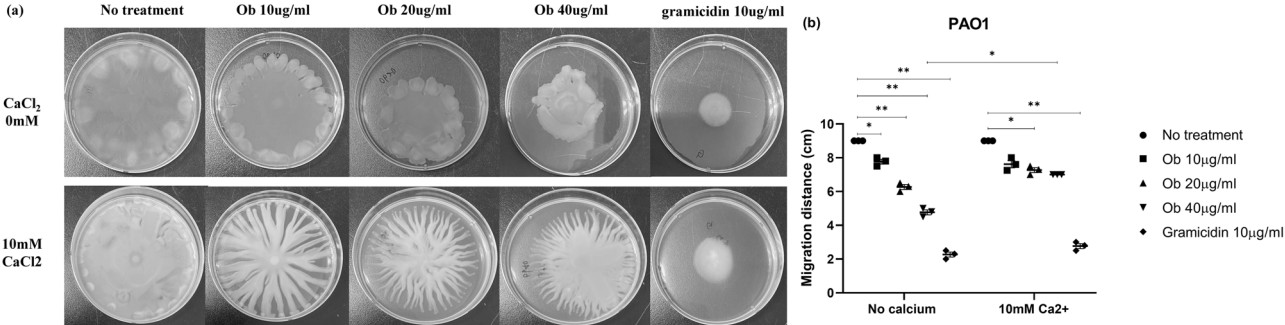

**Fig. 9 Suppression of motility of *P. aeruginosa* strain PAO1 by Ob. a** Motility analysis was performed using semisolid agar plates containing Ob at various concentrations. *P. aeruginosa* strain PAO1 was inoculated into the center of the semisolid agar containing Ob. **b** The migration distance (cm) was measured and recorded after the inoculated plates were incubated at 37 °C. The error bars represent the means and standard deviations (SDs). Statistical analysis was performed using unpaired two-tailed student *t*-test. Data are mean ± SEM for *n* = 3 biologically independent experiment. \**p* < 0.05, \*\**p* < 0.005, \*\*\**p* < 0.0005, \*\*\*\**p* < 0.00005.

combination may be readily developed into a novel therapeutic option for treatment of infections caused by multidrug-resistant Gram-negative pathogens.

## Methods

**Bacterial strains and reagents**. All strains used in this study are listed in Supplementary Table 1. Strains were grown in Luria-Bertani (LB) broth or LB agar plates at 37 °C overnight. All chemicals used in this study were obtained from Sigma-Aldrich.

**Antimicrobial susceptibility tests**. Determination of the minimum inhibitory concentration (MIC) of colistin in the presence and absence of Ob was conducted using a broth dilution method[51]. The results were analyzed according to the Clinical & Laboratory Standards Institute guideline[52]. Briefly, colistin and Ob were two-fold diluted with cation-adjusted MH broth and mixed with equal volume of bacterial suspensions ($1.5 \times 10^6$ colony-forming units (CFUs)/ml) in 96-well microtiter plate. After 16 h incubation at 37 °C, MIC values were determined as the lowest concentration of colistin and Ob that prevents the visible growth of strains. All experiments were performed in triplicate.

To further analyze the synergistic antimicrobial effect of colistin and Ob, a checkerboard assay was performed according to methods described previously[27]. Briefly, Ob and colistin were two-fold serially diluted with 150 µl of MH broth to create an $8 \times 12$ matrix. Bacteria suspension ($2 \times 10^6$ CFUs/ml) were inoculated into the media and incubated at 37 °C for 16 h. The absorbance at 600 nm was determined using SpectraMax ABS Microplate Reader (Molecular Devices, United States). The fractional inhibitory concentration index (FICI) was calculated using the formula as follows: FIC index = (MIC of colistin in combination with Ob)/ (MIC of colistin alone) + (MIC of Ob in combination with colistin)/(MIC of Ob alone). Synergy was defined as an FIC index ≤0.5. All experiments were performed in triplicate.

**Determination of the synergistic antimicrobial effect of the Ob and colistin drug combination**. A time-dependent killing curve was constructed to evaluate the synergistic antimicrobial effect of Ob when it was used in combination with colistin to inhibit colistin susceptible and resistant *E. coli* strains[53]. Briefly, *E. coli* in the exponential phase was treated with a series of concentrations of colistin, Ob, and various combinations of both. Viable cells at each time point were counted in triplicate and killing curves were drawn using GraphPad Prism 8.0 (San Diego, CA, USA).

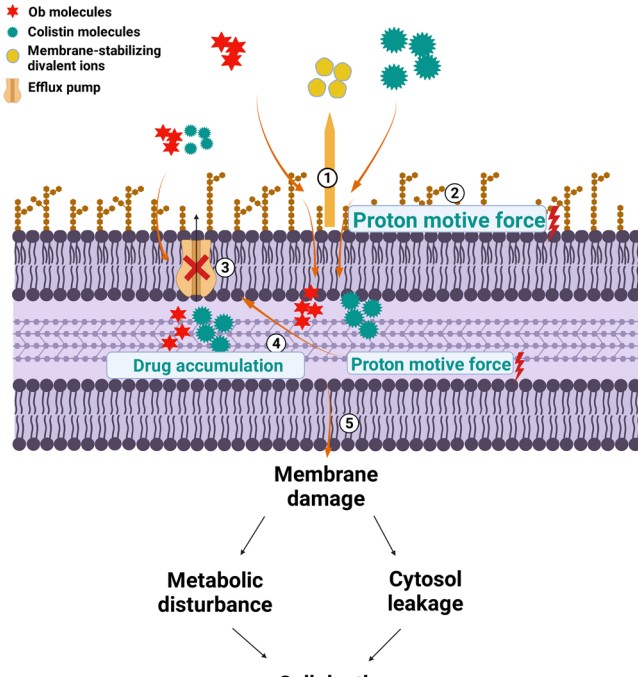

**Fig. 10 Proposed antimicrobial mechanisms of the Ob and colistin combination on colistin-resistant *E. coli*.** Ob acts synergistically with colistin to cause membrane permeabilization by displacing the membrane-stabilizing divalent ions ①. The Ob and colistin combination also causes dissipation PMF ② and inhibits drug efflux ③, resulting in accumulation of the two agents and disruption of electron transport chain activities, which in turn leads to further dissipation of PMF, reduction in efflux activities, and accumulation of the drugs ④. The collapse of PMF and extensive membrane damages result in cytosol leakage and eventually cell death ⑤. The figure was created using BioRender.

**Evaluation of the bactericidal effect of Ob and colistin on colistin-resistant *E. coli* by LIVE/DEAD staining**. The bactericidal effect of the combination of Ob and colistin on colistin-resistant *E. coli* was investigated using the LIVE/DEAD BacLight Bacterial Viability Kit (L1702)[54]. Briefly, *mcr-1*-bearing *E. coli* J53 in the exponential phase was treated with Ob, colistin, and different combinations of both at 37 °C for 2 h. The culture was then centrifuged, and the cell pellet was washed twice with sterile PBS, followed by re-suspension in 300 μL PBS. Each microliter of bacterial suspension was stained with three microliters of dye at room temperature in dark for 15 min. After staining, the bacterial cells were washed twice with PBS and re-suspended in 100 μL PBS. Two microliters of the bacterial suspension were used for imaging by fluorescence microscopy (Nikon Eclipse Ti2 fluorescence microscope, Nikon, Tokyo, Japan). Quantitative analysis was conducted in triplicates by counting the live and dead bacterial cells[54].

**Visualization of the synergistic killing effect of Ob and colistin on colistin-resistant *E. coli* by scanning electronic microscopy (SEM)**. SEM imaging was employed to examine the change in cellular morphology of colistin-resistant *E. coli* incubated with the combination of Ob and colistin according to the procedure described previously, with some modifications[55]. Briefly, MCR-1 producing *E. coli* J53 at the exponential phase was incubated with Ob, colistin, and different combinations of both for 2 h. After treatment, the bacterial cell pellets were washed twice with PBS, followed by fixation with 2.5% glutaraldehyde for 1 h at 4 °C. The fixed cells were then dehydrated with 50% ethanol and 100% ethanol. The cellular morphology was observed using a scanning electron microscope (Tescan VEGA3).

**Cationic ion assays**. The effect of different divalent cations, including $CaCl_2$ and $MgCl_2$, on the antimicrobial effect of the combination of colistin and Ob against colistin susceptible and colistin-resistant *E. coli* was evaluated through determination of MIC of the test agents in the presence of the cations based on the Clinical & Laboratory Standards Institute guideline[56,57].

**Swarming assay**. The effect of Ob on the swarming motility of *Pseudomonas aeruginosa* was investigated as described previously, with some modifications[58]. The medium for the motility assay was tryptic soy broth containing 0.5% of agar

and different concentrations of Ob. 10 μg/ml of gramicidin was also included to compare the effect of Ob. The effect of calcium ions on the swarming phenotype of the strains was also evaluated by the addition of 10 mM $CaCl_2$ during the assay.

**Membrane permeability test**. The bacterial membrane permeability of colistin-resistant and colistin-susceptible *E. coli* was determined using SYTOX Green as described previously[59], with slight modifications. Overnight culture of *mcr-1*-bearing *E. coli* J53 was diluted 100-fold in 3 mL of fresh LB and incubated at 37 °C until the exponential phase was reached. The cultures were then incubated with different concentrations of Ob at 37 °C for 3 h. After incubation, the culture was centrifuged and washed twice with PBS. The pellets were re-suspended in PBS to prepare the bacterial suspension ($OD_{600} = 0.2$). SYTOX Green (1 μM) was added to 1 mL bacterial suspension and cultured in dark for 10 min. The fluorescence intensity of the test samples was monitored using the SpectraMax® iD3 Multi-Mode Microplate Reader (Molecular Devices, Austria) with an excitation and emission wavelength of 488 and 523 nm, respectively.

**Membrane potential assays using 3,3′-dipropylthiadicarbocyanine iodide [DiSC3(5)]**. Bacterial membrane potential was determined using the voltage-sensitive dye [DiSC3(5)] as described previously[27]. Briefly, the stained cells were treated with different concentrations of Ob alone, colistin alone, and different combinations of both. The fluorescence level of the sample was measured for a period of 30 min using a SpectraMax® iD3 Multi-Mode Microplate Reader. 1 μM valinomycin was added as the positive control. The stained bacterial cells were observed using fluorescence microscopy.

**Assessment of efflux inhibitory effect of Ob and colistin using Nile Red**. The inhibitory effect of Ob and colistin on the activities of efflux pumps was evaluated by performing a Nile Red efflux assay[38]. Briefly, bacterial cells ($OD_{600} = 1.0$) treated with different concentrations of colistin, Ob, and various combinations of both were stained with 5 μM Nile red for 3 h at 37 °C. Upon staining, bacterial cells were washed and re-suspended with phosphate-buffered saline (PBS) containing 1 mM $MgCl_2$ and subjected to the indicated treatment. Fluorometric measurements were carried out on black polystyrene microtiter plates for another 2.8 min using a microplate reader, after which the cell suspension was treated with 50 mM glucose. Changes in fluorescence level of the sample were then further monitored for 20 min using a Microplate Reader.

**Mouse infection model**. The antimicrobial effect of the combination of Ob and colistin against colistin-resistant carbapenem-resistant *E. coli* strain CoREC5 was also tested in a mouse infection model according to a method described previously, with some modifications[27]. In this experiment, male NIH mice were divided into four groups (five mice per group) and infected intraperitoneally with $6.0 \times 10^8$ CFU *E. coli* CoREC5. At 1 h post infection (hpi), the mice were then injected intraperitoneally with 8 mg/kg of colistin, 10 mg/kg of Ob, and a combination of both, respectively, every 12 h for 48 h. Four untreated animals were included as control. All animal experiments were approved by the Animal Subjects Ethics Sub-Committee of City University of Hong Kong.

**Statistics and reproducibility**. Statistical analysis was performed using GraphPad Prism 8. All data from at least triplicates are shown as mean ± SD. Unpaired *t* test were used to calculate P values (*$P < 0.05$, **$P < 0.01$, ***$P < 0.001$, and ****$P < 0.0001$).

## Data availability
All data are available within the paper and the Supplementary Information. All data are available from the corresponding authors upon reasonable request. The source data can be found at https://figshare.com/articles/dataset/Ob_numerical_source_data_xlsx/19736755.

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

## Acknowledgements

This research was supported by the Research Impact Fund (R5011-18F) from the Research Grant Council of Hong Kong Government.

## Author contributions

X.C. initiated the project, performed the experiments, and drafted the manuscript; C.Y.L. helped with checkerboard and time-kill analysis; K.C.C. helped with animal experiments; P.Z. helped with SEM imaging; E.W.C.C. helped with experimental design and manuscript writing; S.C. designed the experiments, supervised the project, and wrote the manuscript.

## Competing interests

The authors declare no competing interests.

## Additional information

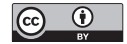 ns license, unless indicated otherwise in a credit line to the material. If material is not included in the article's Creative Commons license and your intended use is not permitted by statutory regulation or exceeds the permitted use, you will need to obtain permission directly from the copyright holder. To view a copy of this license, visit http://creativecommons.org/licenses/by/4.0/.

© The Author(s) 2022

