## [Peer Review File · Communications Biology]

Reviewers' comments:

Reviewer #1 (Remarks to the Author):

In the present work, Xu et al describe the use of the FDA approved drug Otilonium bromide (Ob) as a potential adjuvant for the antimicrobial colistin against gram-negative pathogens. The work described is thorough, timely, and likely to be of high interest to the wider infectious disease community, considering the growing global threat of antimicrobial resistance. The authors do a great job characterizing the combination of Ob and colistin in vitro, in vivo, and with diverse gram-negative bacteria. Overall, I think this study is nicely executed and appropriate for publication at Communications Biology. I only have minor comments for the authors to address in a revision. Well-done!

(1) Figure 1: Please include a scale bar for these plots. Also, the figure would be clearer if the axes were square (row height and column widths equal) and if the x-axis (colistin concentration) increased from left to right (with 0 Ob and 0 colistin at the origin)

(2) Results 3.2, Figure 4: Please explain why CoREC59 are used in these experiments, instead of J53 (w/ or w/o mcr-1). Please indicate in Figure 4 the number of animals used for each treatment arm. Please perform statistical analyses on the CT8+CB10 treatment arm: 4 mice may not be enough to have statistical power to determine if this level of survival is statistically significant.

(3) Figure 8: Please move the microscopy images to the right of the Figure, so that the subpanels are appropriately ordered from (a) to (e).

(4) Results 3.5, Line 299: It is unclear what is meant by "ATP analysis"

(5) Results 3.6, Figure 9: Please explain the use of BW25113 in these experiments. One would expect that J53 lacking mcr-1 plasmid would be the appropriate strain for 9d?

(6) Results 3.7, Figure 11: "Cytosol" is mis-spelled in the figure and "metabolic disturbance" should be capitalized. It is not clear that metabolic changes are directly measured in these experiments. There is extensive evidence that bacterial metabolism contributes to the lethal actions of bactericidal antimicrobials. The authors should consider citing manuscripts by the James Collins lab on this subject if they wish to include metabolism as part of the mechanism for Ob-colistin synergy.

Reviewer #2 (Remarks to the Author):

The manuscript by Xu et al. describes how otilonium bromide synergizes with colistin to sensitize resistant isolates and potentiate bactericidal activity. The manuscript is important and suggests a potential treatment option for colistin resistant bacterial infections. The data are appropriately displayed and interpreted. I have no major concerns. I have listed minor concerns below. The manuscript, although generally well-written contains numerous errors. Some of those are listed below, but there are undoubtedly others. I recommend a careful rewrite of the manuscript to address this.

Minor concerns and writing errors

Line 73: Absorbed by human cells

Line 195: synergistically

Figure 3 could be added as a supplemental figure

Line 222: survival was measured – not eradication

Line 299: ATP analysis?

312: dissipation of membrane dissipation?

I don't understand the section beginning at line 350. It reads like a conclusions section, but that section doesn't actually begin until line 390. As it is still the results section, it needs to be rewritten considerably.

Line 569: The checkerboard measures inhibition, not eradication.

Line 591: sepsis

Figure 11: Why are the drugs apparently entering the cytoplasm? Does that happen? Is it necessary? Wouldn't they exert their effects in the periplasm?

Response to reviewers' comments

Reviewer #1

(1) Figure 1: Please include a scale bar for these plots. Also, the figure would be clearer if the axes were square (row height and column widths equal) and if the x-axis (colistin concentration) increased from left to right (with 0 Ob and 0 colistin at the origin)

Response: We have made revisions accordingly in Figure 1. Scale bar has been added. The x-axis has been changed where the x-axis increases from left to right.

Figure 1. Checkerboard analysis of the synergistic antimicrobial effect of colistin and Ob on both colistin-resistant and colistin-susceptible *E. coli*. (a) Colistin-susceptible *E. coli* J53. (b) Colistin-resistant *E. coli* J53 carrying a *mcr-1*-bearing plasmid which was originally recovered from a clinical *E. coli* strain. CT, colistin.

(2) Results 3.2, Figure 4: Please explain why CoREC59 are used in these experiments, instead of J53 (w/ or w/o *mcr-1*). Please indicate in Figure 4 the number of animals used for each treatment arm. Please perform statistical analyses on the CT8+CB10 treatment arm: 4 mice may not be enough to have statistical power to determine if this level of survival is statistically significant.

Response: CoREC59 exhibited a higher level of virulence than J53 (w/ or w/o *mcr-1*), and it was a clinical strain and more suitable for mouse infection experiments. We actually used 5 mice per group. “4 mice” was a typo at Line 179.

(3) Figure 8: Please move the microscopy images to the right of the Figure, so that the subpanels are appropriately ordered from (a) to (e).

Response: We have made revisions accordingly in Figure 7. We moved the microscopy images to the right of the figure.

Figure 7 Dissipation of membrane potential in *mcr-1*-bearing *E. coli* J53 upon treatment with Ob (a), colistin (b) and various combinations of the two drugs (c). (d) Comparison of effects of different compounds on the bacterial cell membrane potential of *mcr-1*-bearing *E. coli* J53. Valinomycin, a K^+ ionophore, was used as positive control. The assay was conducted in the presence of 100mM of KCl. CT, colistin. (e) Fluorescence microscopic images of bacterial membrane potential of *mcr-1*-bearing *E. coli* J53 taken upon treatment with Ob and colistin. Gramicidin, an ionophore, was used as the positive control.

(4) Results 3.5, Line 299: It is unclear what is meant by "ATP analysis"

Response: It is a typo. It should be ATP synthesis at line 299.

(5) Results 3.6, Figure 9: Please explain the use of BW25113 in these experiments.

One would expect that J53 lacking *mcr-1* plasmid would be the appropriate strain for 9d?

Response: We have made revisions accordingly in Figure 8d. Nile Red efflux of colistin-susceptible *E. coli* J53 treated with Ob, colistin and a combination of both agents was determined using the same method.

Figure 8. Inhibition of Nile Red efflux in *mcr-1*-bearing *E. coli* by Ob (a), colistin (b) and a combination of both agents (c). 50mM of Glucose was added to the assay at 77s and fluorescence was monitored for 3 minutes. (d) Nile Red efflux of colistin-susceptible *E. coli* J53 treated with Ob, colistin and a combination of both agents. 50mM of Glucose was added to the assay at 110s and fluorescence was monitored for 3 minutes.

(6) Results 3.7, Figure 11: "Cytosol" is mis-spelled in the figure and "metabolic disturbance" should be capitalized. It is not clear that metabolic changes are directly measured in these experiments. There is extensive evidence that bacterial metabolism contributes to the lethal actions of bactericidal antimicrobials. The authors should consider citing manuscripts by the James Collins lab on this subject if they wish to include metabolism as part of the mechanism for Ob-colistin synergy.

Response: We have added more discussion as follows "The metabolic status of the bacterial cell is known to be severely impacted by the actions of antimicrobial agents¹ at line 379

Reviewer #2 (Remarks to the Author):

The manuscript by Xu et al. describes how otilonium bromide synergizes with colistin to sensitize resistant isolates and potentiate bactericidal activity. The manuscript is important and suggests a potential treatment option for colistin resistant bacterial

infections. The data are appropriately displayed and interpreted. I have no major concerns. I have listed minor concerns below. The manuscript, although generally well-written contains numerous errors. Some of those are listed below, but there are undoubtedly others. I recommend a careful rewrite of the manuscript to address this.

Minor concerns and writing errors

Line 73: Absorbed by human cells

Response: Revised accordingly at line 73.

Line 195: synergistically

Response: Revised accordingly at line 200.

Figure 3 could be added as a supplemental figure

Response: Revised accordingly in Figure S1. Figure 3 has been added in supplemental materials.

Supplementary Figure S1. LIVE/DEAD Cell Viability Assays depicting the efficacy of Ob and colistin combination therapy *in vitro*. (a) Microscopic observation of *mcr-1*-bearing *E. coli* stained with live and death dye was treated with colistin, Ob and different combinations of both. (b) Quantification of fluorescence signals of random views of 100 cells as presented in (a). Data are representative of three experiments performed in triplicate. *, $p < 0.05$, **, $p < 0.005$, ***, $p < 0.0005$, **** $p < 0.00005$ (independent *t*-test).

Line 222: survival was measured – not eradication

Response: The word “eradicates” was changed to be “eliminates” at line 227.

Line 299: ATP analysis?

Response: It is a typo. It should be ATP synthesis at line 299.

312: dissipation of membrane dissipation?

Response: The words “dissipation of” were erased at line 311.

I don't understand the section beginning at line 350. It reads like a conclusions section, but that section doesn't actually begin until line 390. As it is still the results section, it needs to be rewritten considerably.

Response: This section has been rewritten at line 350.

Line 569: The checkerboard measures inhibition, not eradication.

Response: This section has been rewritten as “Checkerboard analysis of the synergistic antimicrobial effect of colistin and Ob on both colistin-resistant and colistin-susceptible *E. coli*” at line 568

Line 591: sepsis

Response: Revised accordingly at line 580.

Figure 11: Why are the drugs apparently entering the cytoplasm? Does that happen? Is it necessary? Wouldn't they exert their effects in the periplasm?

Response: Since the premetalized bacterial cell membrane caused by the combination of Ob and colistin allowed the uptake of colistin, colistin could interact with liposaccharide of cytoplasmic membrane². Ob is an amphipathic compound which enable it to partition into the phospholipid bilayer of membrane^{3,4}. We are not sure whether the drugs would enter the cytoplasm. To avoid misunderstanding, we revised the Figure 10 and remove the symbol of Ob and colistin at the bottom of figure..

Figure 10. Proposed antimicrobial mechanisms of the Ob and colistin combination on colistin-resistant *E. coli*. Ob acts synergistically with colistin to cause membrane permeabilization by displacing the membrane-stabilizing divalent ions ①. The Ob and colistin combination also causes dissipation PMF ② and inhibits drug efflux③, resulting in accumulation of the two agents and disruption of electron transport chain activities, which in turn leads to further dissipation of PMF, reduction in efflux activities and accumulation of the drugs ④. The collapse of PMF and extensive membrane damages result in cytosol leakage and eventually cell death ⑤. The figure was created using BioRender.

References:

- 1 Lopatkin, A. J. *et al.* Bacterial metabolic state more accurately predicts antibiotic lethality than growth rate. *Nature microbiology* 4, 2109-2117 (2019).
- 2 Trimble, M. J., Mlynářčík, P., Kolář, M. & Hancock, R. E. Polymyxin: alternative mechanisms of action and resistance. *Cold Spring Harbor*

perspectives in medicine **6**, a025288 (2016).

- 3 Andersen, C., Holland, I. & Jacq, A. Verapamil, a Ca²⁺ channel inhibitor acts as a local anesthetic and induces the sigma E dependent extra-cytoplasmic stress response in *E. coli*. *Biochimica et Biophysica Acta (BBA)-Biomembranes* **1758**, 1587-1595 (2006).
- 4 Shi, B. & Tien, H. T. Action of calcium channel and beta-adrenergic blocking agents in bilayer lipid membranes. *Biochimica et Biophysica Acta (BBA)-Biomembranes* **859**, 125-134 (1986).